# Antibiosis Effects of Rice Carrying *Bph14* and *Bph15* on the Brown Planthopper, *Nilaparvata lugens*

**Liangmiao Qiu** [1,2], **Wuqi Wang** [2], **Longqing Shi** [2] , **Qiquan Liu** [1,2] **and Zhixiong Zhan** [1,2,3,*]

[1]   Institute of Plant Protection, Fujian Academy of Agricultural Sciences, Fuzhou 350013, China;
     bjndqlm@163.com (L.Q.); liuqq1221@163.com (Q.L.)
[2]   Fujian Key Laboratory for Monitoring and Integrated Management of Crop Pest, Fuzhou 350013, China;
     ricewwq@163.com (W.W.); shi618156@sina.com (L.S.)
[3]   Rice Research Institute, Fujian Academy of Agricultural Sciences, Fuzhou 350019, China
[*]   Correspondence: zzx64@sohu.com; Tel.: +189-5048-9648

**Abstract:** The brown planthopper(BPH), *Nilaparvata lugens*, is the most destructive insect pest in rice production worldwide. The development and cultivation of BPH-resistant varieties is the most economical and efficient strategy to overcome the destruction caused by BPH. The modified bulk seedling test method was used to identify the BPH resistance level and host feeding choice of rice lines of Liangyou8676(*Bph14/Bph15*), Bph68S(*Bph14/Bph15*), RHT(*Bph3*), Fuhui676, and TN1 on BPH. In the meantime, the population, survival and emergence rate, developmental duration, honeydew excretion, female ratio and brachypterous ratio of adults were used as indicators to detect the antibiosis effects of the different rice lines. The results showed that the resistance levels of RHT, Bph68S, Liangyou8676, Fuhui676, and TN1 to BPH were HR, R, MR, S and HS, respectively. The host choice implied that BPH was more inclined to feeding on rice plants with a lower resistance. An analysis of the antibiosis activity of rice lines RHT, BPh68S, and Liangyou8676 carrying resistance genes indicated a significant reduction in the population growth rate, survival and emergence rate of BPH nymphs, significant delay in the developmental duration of nymphs, reduced honeydew excretion of females, decreased female ratio, and a decreased brachypterous ratio of females and males, when compared with rice carrying no BPH-resistant genes.

**Keywords:** antibiosis; brown planthopper; *Bph14*; *Bph15*; resistance gene

## 1. Introduction

Rice (*Oryza sativa* L.) is the staple food crop for more than three billion people worldwide. The brown planthopper (BPH) is the most destructive insect pest of rice [1–4]. BPH extracts the phloem sap of rice plants using its piercing and sucking mouthparts. Light BPH infestation affects the growth of rice plants; whereas heavy infestation results in "hopper burns." [1]. BPH also serves as a vector that transmits the ragged stunt virus and the gassy stunt virus [2,3]. In recent years, BPH infestations have intensified across Asia, causing heavy yield losses in rice production [4–6]. Therefore, controlling BPH is a challenging food safety issue. Traditionally, farmers solely depended on chemical pesticides for controlling BPH [7,8]. However, the overuse of pesticides has resulted in a series of negative effects such as resistance of pests to synthetic chemicals, pests resurgence, and environmental contamination [9–11].

The use of resistant rice is the most economical, efficient, and environment-friendly measure for controlling BPH [12–14]. Therefore, identifying resistance genes and growing resistant rice cultivars have great significance in the integrative pest management of BPH [15,16]. Lines showing BPH-resistant genes are abundant in rice germplasm collections worldwide [17,18]. Rice lines that are resistant to

BPH are representative of a long history of breeding, with a successful application in rice production. For example, IR26, the first commercial resistant rice variety carrying *Bph1*was released in 1973. Resistance to BPH has played a major role in the integrated control strategies for the pest throughout Asia [19–23]. Up to now, at least 34 BPH-resistance genes have been reported. Among them, 19 and 15 are dominant and recessive, respectively; 28 major resistance genes have been mapped and 9 genes have been cloned [24].

Plants can perform various resistance ecological mechanisms to reduce pest damage in nature. Plant resistance to pests is generally differentiated in (1) antibiosis: a characteristic that reduces pest survival, growth rate, or reproduction following the ingestion of host tissue; (2) antixenosis: a characteristic that repels or disturbs pests, causing a reduction in feeding, colonization or oviposition; and (3) tolerance: a capacity to produce a crop of high yield and quality despite of pest infestation [10,14,25]. However, the information on resistance mechanism of the BPH resistance genes is limited. Therefore, it is necessary to evaluate the resistance level and analyze its mechanisms of resistance in resistant cultivars carrying BPH resistance genes, which should favor for the BPH resistance breeding programs in rice.

*Bph14* and *Bph15* are two major BPH resistance genes that have been mapped in B5 [25,26], derived from wild rice *Oryza officinalis* Wall ex Watt. The resistance gene of *Bph14* has been cloned and its resistance mechanisms had also been identified [10]. Now, resistance genes of *Bph14* and *Bph15*are been applied extensively in rice breeding through molecular marker-assisted selection (MAS) [27–29].

Bph68S is a newly bred two-line sterile rice carrying resistance genes of *Bph14* and *Bph15*derived from high BPH-resistance rice of B5,by the strategy of molecular marker-assisted selection(MAS), and it presents excellent agronomic characters, such as completed male sterility, exerted stigma for better outcrossing and resistance to BPH, in the meantime, using the Bph68S as paternal parent, the rice combination of Liangyou234 was bred using and presented good resistance to BPH [30]. So the Bph68S has a good application prospect in hybrid breeding of rice. Although the stable rice line of Bph68S is expected to apply in BPH control, this has yet to be further demonstrated for the potential incorporation of the line in further rice breeding.

Fuhui676 is an excellent indica restorer line which was bred by us, and does not carry any known BPH resistance gene. Using Bph68S and Fuhui676, we successfully bred a new hybrid combination of Liangyou8676, which carries resistance genes of *Bph14* and *Bph15*, and exhibited excellent agronomic characteristics and resistance to BPH in field. For identification the BPH resistance of rice carrying *Bph14* and *Bph15* derived from B5, an evaluation of BPH resistance and antibiosis resistance of Bph68S and Liangyou8676 to BPH was carried out in laboratory. The present study is of significance for future BPH resistance evaluations and for the further breeding of rice lines to achieve resistance to BPH, and it will provide scientific evidence for the sustainable control of BPH using resistant rice.

## 2. Materials and Methods

### 2.1. Rice Plants and BPHs

Two-line sterile rice Bph68S, carrying resistance genes *Bph14* and *Bph15*, resistant to BPH biotypes 1, 2, and 3 was used. The indica restorer line of Fuhui676 does not carry any known resistance gene to BPH. A combination of Lianyou8676, carrying resistance genes *Bph14* and *Bph15*, was obtained by the hybridization of the parents of Bph68S and Fuhui676. TN1 does not carry any BPH resistance gene, and, therefore, is highly susceptible to BPH. It has been used as the susceptible control in this study. RHT(RathuHeenati), carrying BPH resistance gene *Bph3*, resistant to all four BPH biotypes [19,31], was used as the resistance control in this study.

The BPHs were collected from rice fields during August 2016 from Fuzhou (26°16′ N, 119°15′ E; where BPH populations of biotype-2 dominated), Fujian province, China. BPHs were placed on TN1 plants in a growth room at the Plant Protection Research Institute of Fujian Academy of Agricultural Sciences, for a 14:10 h light: dark photoperiod; the temperature was maintained at 25 ± 1 °C and the relative humidity was around 75%. The insect-rearing protocol was followed for a year.

## 2.2. Evaluation of Resistance and Host Choice of Rice to BPH

For the identification and evaluation of the BPH resistance of rice, the modified bulk seedling test was carried out as described by IRRI (1988) [32] and Huang et al. (2001) [25]. Rice seeds were soaked in water and germinated at 30 °C. The germinated seeds of each rice line were sown using the method of randomized complete block, at a planting density of 2.5 cm between plants in a row and 4 cm between rows in a plastic tray (45 × 30 × 9 cm) containing pulverized soil. The seedlings were thinned at the three-leaf stage to 10 plants of each line and infested with the second instar nymphs of BPHs at a density of 8 insects per seedling and then covered with fine, light-transmitting cage. The bioassay experiment was replicated three times and carried out in the growth room that maintained the BPH. To evaluate the BPH host feeding choice, the number of BPHs that settled on different rice seedlings was calculated and calculated after 1, 2, 3, and 4 days of infestation. The target seedlings were then examined, and each seedling was given a score of 0, 1, 3, 5, 7 or 9 after all of the TN1 plants had died (ca. 9–10 days after infestation). Here, the 0 score indicated no damage of rice seedling, whereas 9 indicated all the seedlingwilted or dead. Accordingly, 1, 3, 5, and 7 indicated 1 leaf yellowing, 1–2 leaves yellowing, 1 leaf wilted and 2–4 leaves wilted, respectively. The criteria for scoring were adapted from IRRI (1988) [32] and Huang et al. (2001) [25]. Lower scores indicate higher resistance to BPH. The average resistance score of each line was calculated as the weighted average of the scores for all the seedlings tested. Based on the resistance level evaluation standard of Hu et al. (2016) [29], the rice lines were classified as highly resistant(HR), resistant(R), moderately resistant(MR), moderately susceptible(MS), susceptible(S), and highly susceptible(HS) corresponding to 0–0.9, 1.0–2.9, 3.0–4.9, 5.0–6.9, 7.0–8.9, and 9, respectively.

## 2.3. Determination of Population Growth Rate, Survival Rate, and Emergence Rate of BPHs

The population growth rate and survival rate of BPH on different rice lines were performed based on the method described by Qiu et al. (2010) [33]. Six equally big seedlings at the three-leaf stage were selected and transferred to a plastic cup (5.5 cm bottom diameter, 8.5 cm opening diameter, 15 cm height) containing the Hoagland solution. The solution was replaced every three days but was always maintained at a level good enough to submerge the root during the experiment. To measure the BPH population growth rate on rice seedlings, each cup was infested with six second instar nymphs which were previously weighed, and two cups/seedlings were taken as an experimental treatment. Six days after the treatment, the weight of the surviving BPHs was recorded using a 0.1 mg sensitivity balance. The population growth rate of the surviving BPHs was calculated based on the method described by Klingler et al. (2005) [34].

To examine the survival rate of BPHs on the rice seedlings, each cup was infested with six first-instar nymphs that hatched out within 12 h; the number of BPHs that survived in each cup was recorded on the ninth days, after the BPH nymphs were released into the cups. The hoppers were placed on fresh seedlings every three days in the same cup as mentioned above during the experiment. To detect the emergence rate, the surviving BPHs were continuously observed until adults emerged. The survival and emergence rates were calculated as the percentage of surviving and emerging hoppers divided by the total number of nymphs released into the cups at the start of the experiment.

The evaluation of the BPH population growth rate, survival rate, and emergence rate on each line or line seedlings was conducted for eight replicates in the same growth room that maintained the BPH.

## 2.4. Individual Development Duration of Nymphs on Rice Plants

To determinate the individual development duration of nymphs, the neonates of BPHs that hatched out within 4 h were collected, and each of them was reared on rice seedlings in the same cup as mentioned above. The BPH development was detected and recorded every 24 h until the adults emerged. When adults emerged, the duration of the development of nymphs (i.e., the time needed for

the neonate develop into adult) were recorded. Sixteen replicates were tested for each rice line under the same environmental conditions of the growth room described previously.

### 2.5. Quantification of Honeydew Excretion of Female Adults

Honeydew collection was carried out as described by Pathak et al. (1982) [35]. The emerging brachypterous females (12–24 h old), which previously starved for 3 h, were transferred individually into a parafilm sachet (3.5 × 3.5 cm) and attached to the main tiller of a 40-day-old rice plant 10 cm above base of rice stem. After 24 h feeding, the weight of the honeydew in each sachet was measured using a 0.1-mg sensitivity balance per BPH female. A total of 8 replicates were tested for each rice line in the same growth room described previously.

### 2.6. Wing Formations, Genders and Brachypterous Ratio of BPH Adults

For detecting the different line of rice affected on the BPH adult wing formations, genders and brachypterous ratio. Each germinated seed from all lines were grown as described above in individual plastic trays covered with a light-transmitting cage. After sowing for 40 days, 120 first-instar nymphs of BPH were collected and released into the cage. After feeding for 15 days, all of the BPH adults were collected, and the female ratio, and the brachypterous ratio of males and females were recorded. The experiment was carried out for three replicates on each rice line in a greenhouse under natural light at a temperature range of 25°C–35°C between September and October 2017.

### 2.7. Statistical Analysis

Statistical analysis of the data was carried out using SPSS18.0(IBM SPSS STATISTICS 18.0, Chicago, USA), one-way ANOVA and the significant difference was identified by the LSD (least significant difference) test at 5% probability level. Percentage values were arcsine transformed prior to analysis.

## 3. Results

### 3.1. BPH Resistance Level

The severity scores of TN1, Fuhui676, Liangyou8676, Bph68S, and RHT against BPH were 9.0, 8.5, 4.1, 2.8 and 0.4, respectively (Figure 1), which showed that the resistance levels of RHT, Bph68S, Liangyou8676, Fuhui676, and TN1 to BPH were HR, R, MR, S, and HS, respectively. This indicated that the resistance of RHT(*Bph3*) to BPH was higher than that of Bph68S(*Bph14/Bph15*), whose resistance level was, in turn, higher than that of Liangyou8676(*Bph14/Bph15*), Fuhui676 (no BPH-resistance gene), and TN1(no BPH-resistance gene).

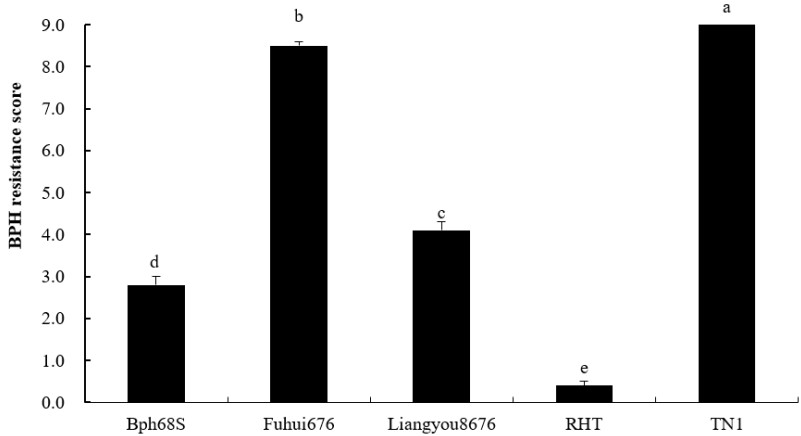

**Figure 1.** BPH resistance scores of rice lines. Error bars indicate standard deviation. And bars labeled with different letters indicate significant difference (*p* < 0.05, one-way ANOVA, LSD multiple comparison).

### 3.2. Host Feeding Choice of BPH Rice to Lines

The results of BPH host feeding choice showed that there was an obvious difference in the number of BPH nymphs that settled on the rice seedlings from 1 to 4 days of infestation (Figure 2). TN1 had the largest number of BPHs settled on, followed by Fuhui676, followed by Liangyou8676, Bph68S, and RHT. The number of BPHs settled on the plants of Bph68S, Liangyou8676, and RHT after 1, 2, 3, and 4 days of infestation was significantly less than those settled on TN1 and Fuhui676. The host feeding choice tests implied that BPH exhibited more inclination to feeding on rice plants with a lower resistance.

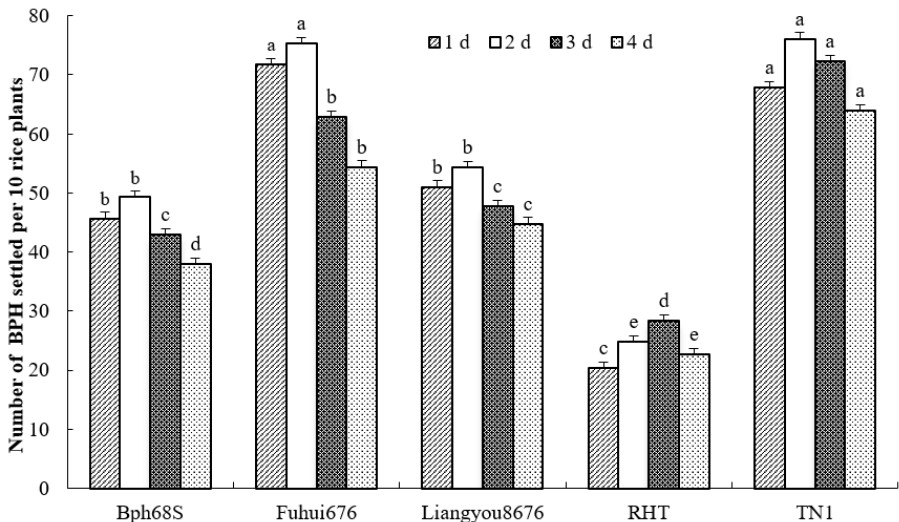

**Figure 2.** Host feeding choice of BPH on rice. Bar graph represent the number of BPHs settled on seedlings of different rice lines. Error bars indicate standard deviation. And bars labeled with different letters indicate the significant difference ($p < 0.05$, one-way ANOVA, LSD multiple comparison).

### 3.3. Population Growth Rate, Survival Rate, Emergence Rate, and Developmental Duration of BPH Nymphs

The influence of the rice lines on the population growth rate, survival rate, emergence rates, and developmental duration of BPHs was investigated and determined (Table 1). The population growth rates of BPHs on TN1 and Fuhui676 were 0.28 and 0.27 mg/BPH/d, respectively, which were significantly larger than the population growth rates of BPHs on the trial rice lines of Bph68S, Liangyou8676, and RHT, carrying BPH resistance genes. This means that the BPHs grow faster feeding on rice carrying no resistance genes. There was no significant difference between the population growth rates of BPHs on Bph68S and Liangyou8676; however, the population growth rate of BPHs on RHT (carrying resistance gene *Bph3*) was significantly smaller than that of Bph68S and Liangyou8676 carrying resistance genes *Bph14/Bph15*.

**Table 1.** The population growth, survival, emergence rates and developmental duration of nymphs on different rice lines.

| Rice Line | Population Growth Rate (mg/BPH/d) | Survival Rate (%) | Emergence Rate (%) | Developmental Duration (d) |
|---|---|---|---|---|
| Bph68S | 0.19 ± 0.022 b | 47.92 ± 5.89 d | 43.75 ± 3.86 d | 15.1 ± 0.9 ab |
| Fuhui676 | 0.27 ± 0.018 a | 72.91 ± 5.89 b | 63.54 ± 4.31 b | 13.8 ± 0.7 c |
| Liangyou8676 | 0.21 ± 0.019 b | 58.33 ± 6.30 c | 51.04 ± 6.95 c | 14.7 ± 0.9 b |
| RHT | 0.15 ± 0.020 c | 40.63 ± 5.34 e | 34.38 ± 5.34 e | 15.6 ± 0.9 a |
| TN1 | 0.28 ± 0.021 a | 82.29 ± 5.34 a | 77.08 ± 5.89 a | 13.4 ± 1.1 c |

Data in the table are Mean ± SD. Means in the same column followed by different letters indicate significant difference ($p < 0.05$, one-way ANOVA, LSD multiple comparison).

There were significant differences in the survival and emergence rates among different rice lines. For example, the survival and emergence rates of the BPH nymphs on TN1 were 82.29% and 77.08%, respectively, which were significantly higher than those on the other four rice lines (approximately 40.63–72.92% and 34.38–63.54%, respectively). Compared with the rice that susceptible to BPH of TN1 and Fuhui676, the survival and emergence rates were significantly reduced by the resistance-BPH rice, with the influence of RHT(*Bph3*) being greater than that of Bph68S and Fuhui676 carrying genes of *Bph14* and *Bph15*.

There was a significant difference in the developmental duration of nymphs on rice lines (TN1 and Fuhui676) that were susceptible to BPH and those (Bph68S, Liangyou8676, and RHT) that were resistant to BPH. The developmental duration of nymphs on TN1 and Fuhui676 was 13.4 and 13.8 days, respectively, whereas that of nymphs on Bph68S, Liangyou8676, and RHT was 15.1, 14.7, and 15.6 days, respectively. This implied that the developmental duration of BPH nymphs was obviously retarded by resistant rice, but the effect of being retarded by RHT(*Bph3*) was greater than that of being retarded by Bph68S(*Bph14/Bph15*) and Liangyou8676(*Bph14/Bph15*).

### 3.4. Effect on Honeydew Excretion, Female Ratio, and Brachypterous Ratio of BPH Adults

Honeydew excretion, female ratio, and brachypterous ratio of BPH adults were detected and calculated (Table 2). The results showed that the honeydew excretion on rice lines (TN1 and Fuhui676) susceptible to BPHs was significantly larger than on rice lines (Bph68S, Liangyou8676, and RHT) resistant to BPH. For example, the honeydew excretion on RHT was the least and that on TN1 was the largest. The values of both were significantly different from those of the other three rice lines. However, there was no significant difference between the values of Bph68S and Liangyou8676.

**Table 2.** Honeydew excretion, female ratio, and brachypterous ratio of BPH adults on different rice lines.

| Rice Line | Honeydew (mg/BPH/d) | Female Ratio (%) | Brachypterous Ratio (%) | |
|---|---|---|---|---|
| | | | Female | Male |
| Bph68S | 5.43 ± 0.84 c | 43.68 ± 2.51 c | 76.97 ±4.45 c | 3.20 ±0.15 c |
| Fuhui676 | 17.49 ± 1.80 b | 54.78 ± 2.75 a | 93.78 ± 3.46 a | 6.81 ± 1.23 b |
| Liangyou8676 | 6.14 ± 0.42 c | 48.70 ± 1.21 b | 85.45 ± 1.15 b | 3.95 ± 1.67 c |
| RHT | 1.09 ± 0.44 d | 37.60 ± 2.36 d | 63.65 ± 2.81 d | 0.00 ± 0.00 d |
| TN1 | 22.44 ± 2.84 a | 56.74 ± 2.81 a | 95.39 ± 0.86 a | 10.64 ± 1.46 a |

Data in the table are Mean ± SD. Means in the same column followed by different letters indicate significant difference ($p < 0.05$, one-way ANOVA, LSD multiple comparison).

The female and brachypterous female ratios of BPH adults on different rice lines ranged from37.60% to 56.74% and 63.65% to 95.39%, respectively. This indicated that BPHs exhibited significantly higher female and brachypterous female ratios on rice lines susceptible to them when compared with of resistance to BPH did, there was no significant difference between TN1 and Fuhui676, but there was significant difference among the rice lines of Bph68S, Liangyou8676 and RHT, the female and brachypterous female ratios on RHT were lowest, then on Bph68S, followed by Liangyou8676.

The brachypterous ratio of males on the different rice lines was very low, ranging between 0 and 10.64%. The ratio on resistant rice lines Bph68S and Liangyou8676 was significantly less than that of TN1 and Fuhui676; there was no brachypterous ratio of males on RHT in this study.

## 4. Discussion

The BPH resistance level and host feeding choice of different rice lines were detected using a modified bulk seedling test. The results indicated that the severity scores of rice lines were RHT<BPh68S<Lianyou8676<Fuhui676< TN1 (Figure 1). RHT scored 0.4 in this study, exhibited a high resistance (HR) to BPH, and fitted well with the findings of Li et al. (2011) [36] and Hu et al. (2016) [29], but differed from that of moderately resistant reported by Qiu et al. (2011) [37] and Chen et al.

(2016) [38], it may be related to the difference of BPH populations characteristic in different province of China. Bph68S (*Bph14* and *Bph15*) was derived from B5 through marked-assisted selection (MAS), was resistant to BPH, with a severity score of 2.8, consistent with the rice line B5(carrying *Bph14* and *Bph15*) with scores<3.0 [3,27,36,37]. In the meantime, the host choice was evaluated, which indicated that there was an obvious difference in the numbers of BPHs that settled on the rice seedlings of different resistance levels. The number of BPHs that settled on rice seedlings carrying resistance genes (*Bph3*, *Bph14*, and *Bph15*) was significantly less than that settled on seedlings carrying no resistance genes (Figure 2). This implied that BPHs preferred to feed on susceptible rice plants carrying no resistance genes; the results coincided well with the research of Li et al. (2011) [36] and Han et al. (2018) [39].

In this study, the indicators, population growth rate, survival rate, emergence rate, developmental duration, honeydew excretion, female ratio, and brachypterous ratio of adults were used to detect the antibiosis to BPHs in four different rice lines using TN1 as the experimental control. The results showed that rice lines Bph68S, Liangyou8676, RHT, and Fuhui676 had antibiosis effects on BPHs when compared with those on TN1. Resistant rice lines Bph68S, Liangyou8676, and RHT significantly inhibited the population growth rate, survival rate, and emergence rate of the BPH nymphs, significantly delayed the developmental duration of the nymphs (Table 1), reduced the honeydew excretion of females, and decreased the female ratio, and brachypterous ratio of females and males (Table 2). The results indicated that rice lines carrying resistance genes *Bph3*, *Bph14*, and *Bph15* had significant antibiosis effects on BPHs, which coincided well with the results of Li et al. (2011) [36] and Hu et al. (2012) [27], however, the antibiosis indicators of rice lines (Bph68S and Liangyou8676) carrying resistance genes*Bph14*/*Bph15* were significantly weaker than that of RHT(carrying *Bph3*),it differed from the gene effect of *Bph14*/*Bph15* (B5)>*Bph3*(RHT) reported by Qiu et al. (2011) [37] and Jiang et al. (2018) [3], The observation may be related to the difference in the expression of *Bph14*/*Bph15* in rice lines Bph68S, Liangyou8676, and B5, the different rice lines carrying *Bph14*/*Bph15* genes performed different resistance level to BPH in the study of Hu et al. (2012) [27]. In addition, though the susceptible rice line Fuhui676 carried no resistance genes, it had significant inhibition effects on the population growth rate, survival rate, and emergence rate, and reduced the honeydew excretion and brachypterous ratio of males; however, there was no significant difference in the developmental duration, female ratio, and the brachypterous ratio of females of Fuhui676 and TN1, which suggested the possibility of other mechanisms contributing to the antibiosis effects to BPHs in rice carrying no BPH resistance gene.

At present, the research of BPH resistance gene mainly focused on gene mapping. Up to now, there are 28 major resistance genes have been mapped and 9 genes have been cloned [24]. But only *Bph14* has been studied deeply in the mechanism of resistance. Du B et al. [10] Identified the mechanisms of *Bph14*, it showed that *Bph14* encodes a coiled-coil, nucleotide-binding, and leucine-rich repeat protein. And the expression of *Bph14* activates the salicylic acid signaling pathway and induces callose deposition in phloem cells and trypsin inhibitor production after planthopper infestation, thus reducing the feeding, growth rate, and longevity of the BPH insects.Although the main mechanisms of BPH resistance genes are not clear now. But we speculate the resistance gene can activate the resistance response of rice after being infested by BPH. In the future, we will focus on the study of mechanism of BPH related resistance genes. And we will try to polymerizethe resistance genes of *Bph3*, *Bph14* and *Bph15*, aims to improve the resistance of rice to BPH.

Presently, there is approximately $1.67 \times 10^7$ hm$^2$ of land area devoted annually to the cultivation of hybrid rice in China, with the hybrid rice accounting for 65% of the total rice yield. In the meantime, there is approximately $0.64 \times 10^7$ hm$^2$ of area devoted to the cultivation of hybrid rice in other countries [40,41]. Therefore, high quality and yield of resistant hybrid rice is crucial ensuring food security worldwide. The BPH is the most destructive insect pest of rice; it is also a major threat to rice production worldwide, especially Asia. Now, it is common knowledge that the most economical, efficient, and environment-friendly measure to control BPH is to cultivate resistant rice. Consequently, breeding resistant varieties suitable for large-scale production is of significance in meeting the needs of rice production worldwide.

## 5. Conclusions

The resistance level and antibiosis effects of rice on BPH were detected in this study. The resistance level and antibiosis effects of the different rice lines were not only determined by BPH resistance genes, but the rice lines carrying resistance genes(*Bph3*,*Bph14/Bph15*) had a significantly stronger antibiosis on BPHs than that having none. It manifested that the BPH resistance level was mainly determined by antibiosis, and that the antibiosis effects were mainly conferred by the resistance genes. So, the population growth rate, survival rate, emergence rate, developmental duration, honeydew excretion, female ratio, and brachypterous ratio could be effective indicators in the evaluation of the antibiosis of resistance rice.

**Author Contributions:** Methodology, conceptualization and funding acquisition, Z.Z.; resources of rice seed, W.W.; formal analysis, L.Q.; investigation, L.Q., Q.L., and L.S.; data curation, L.Q., L.S.; writing—original draft, L.Q., L.S., Q.L.; writing—review and editing, L.Q., L.S., Q.L., Z.Z.; funding acquisition, Z.Z. All authors have read and agreed to the published version of the manuscript.

**Funding:** This research was funded by Ministry of Science and Technology of China, grant No. 2017YFD0100100, the Science and Technology Innovation Team of Fujian Academy of Agricultural Sciences, grant No. STIT2017-1-8 and the Doctoral Project of Fujian Academy of Agricultural Sciences, grant No. DC2017-8.

**Acknowledgments:** We express our gratefulness to the anonymous reviewers for their valuable comments on the earlier versions of the manuscript.

**Conflicts of Interest:** The authors declare no conflict of interest.

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
