# Peer review of "Antibiosis Effects of Rice Carrying Bph14 and Bph15 on the Brown Planthopper, Nilaparvata lugens"

_agriculture, doi:10.3390/agriculture10040109_

Round 1

Reviewer 1 Report

-Lines 29-30. Include citation(s) supporting the claim that BPH is the most destructive insect pest of rice.

-In section 2.2, describe the experimental design with more detail. For example, what design was used (e.g. randomized complete block)? How many replications/environments?

-In the discussion section, please speculate on the mechanisms underlying these resistance genes. For example, have previous studies identified proteins, pathways, metabolites, morphological characteristics, etc. associated with these two resistance genes or related resistance genes?

Author Response

Comment 1: -Lines 29-30. Include citation(s) supporting the claim that BPH is the most destructive insect pest of rice.

Response 1: The references of 1-4 all claimed that the BPH is the most destructive insect pest of rice. So the citations supporting the claim have replenished in the revised manuscript.

Comment 2: -In section 2.2, describe the experimental design with more detail. For example, what design was used (e.g. randomized complete block)? How many replications/environments?

Response 2: In section 2.2, the randomized complete block was designed in the experiment. The replications and environments have been described in the manuscript. Please see the revised manuscript for details.

Comment 3: -In the discussion section, please speculate on the mechanisms underlying these resistance genes. For example, have previous studies identified proteins, pathways, metabolites, morphological characteristics, etc. associated with these two resistance genes or related resistance genes?

Response 3: Thanks for your kind advices. We did some revises in the discussion section of mechanisms underlying these resistance genes. Please see the revised manuscript for details.

Reviewer 2 Report

The authors describe a large experiment where they tested the antibiosis effects of 4 rice lines for insect pest Brown plant hopper or BPH.

The methods and the experimental design are well described and scientifically sound, although I would have appreciated a bit more rigor in the statistical analysis, e.g. use of box plots in the figures instead of box-whisker plots, and omission of presentation of individual replicate experimental data in favor of experimental averages.

Additionally-although it is mentioned that at least one rice line(Liangyou8676) is a hybrid of another(Bph68S) included in the study, influence of such genetic relationships was not explored during analysis.

I would have appreciated a little more background and definition for antibiosis in the introduction section to develop and clarify the hypothesis being tested. As presented- the hypotheses tested is not clear- my understanding is

1) There is a difference in antibiosis between major resistance gene (Bph14, Bph15 or Bph3) containing rice lines vs those that don't

2) Among resistance gene containing lines, is Bph14/15 stack more or less efficient than Bph3?

3) Is there a difference between antibiosis between inbred lines versus their hybrids?

There may be others the authors intended to present- but that is all I can infer based on what is presented. Clarification of these hypotheses by the authors would be preferable.

Another thing I didn't exactly understand from the presentation is the significance of female nymph versus male nymphs? What are the prior findings in the literature about this observation in relevance to antibiosis? Why is it important? More clarification would be welcome.

The authors mentioned that at least one of the genes being tested as a source of resistance was isolated. A short review of what the proposed genetic mode of action for these resistance genes being tested would be welcome.

For the writing style- I have noticed some interesting uses of "meanwhile" in the text. I would suggest that the authors review the use of the word in the text.

Finally- as the authors discovered that the Bph3 may have a stronger antibiosis effect on BPH then Bph14/15 stack- I would have liked to see this spelled out and maybe a short discussion of what they plan to do next- Future work?

Author Response

Comment 1: The authors describe a large experiment where they tested the antibiosis effects of 4 rice lines for insect pest Brown plant hopper or BPH.

Response 1: Thank you so much for your high opinion of our experiment.

Comment 2: The methods and the experimental design are well described and scientifically sound, although I would have appreciated a bit more rigor in the statistical analysis, e.g. use of box plots in the figures instead of box-whisker plots, and omission of presentation of individual replicate experimental data in favor of experimental averages.

Response 2: In order to reflect the difference of antibiosis effects between different rice lines more directly, so we use the box-whisker plots in the figures.The average resistance score of each line was calculated as the weighted average in favor of experimental averages.

Comment 3: Additionally-although it is mentioned that at least one rice line(Liangyou8676) is a hybrid of another(Bph68S) included in the study, influence of such genetic relationships was not explored during analysis.

Response3: We have identified that the line of Liangyou8676 carries resistance genes of Bph14/Bph15 in our prior studies, and we also noted that the Liangyou8676 carries gene of Bph14/Bph15 derived from Bph68S in our manuscript. As for the influence of genetic relationships, we will study it further in future.

Comment 4: I would have appreciated a little more background and definition for antibiosis in the introduction section to develop and clarify the hypothesis being tested. As presented- the hypotheses tested is not clear- my understanding is

Response4: We have filled out the background and definition for antibiosis in the introduction section in revised manuscript. Please see the revised manuscript for details.

1) There is a difference in antibiosis between major resistance gene (Bph14, Bph15 or Bph3) containing rice lines vs those that don't

Response 1): In contrast to the results of other studies, we have put it in the discussion section in manuscript.

2) Among resistance gene containing lines, is Bph14/15 stack more or less efficient than Bph3?

Response 2): According to our results, Bph14/15 in Bph68S and Liangyou8676 is less efficient than Bph3 in RHT.

3) Is there a difference between antibiosis between inbred lines versus their hybrids?

Response 3): There is a difference between antibiosis between inbred lines versus their hybrids. Our results showed that the antibiosis of hybrid line(Liangyou8676) is less efficient than inbred line (Bph68S).

Comment 5: There may be others the authors intended to present- but that is all I can infer based on what is presented. Clarification of these hypotheses by the authors would be preferable.

Response 5: These hypotheses are based on that the rice lines of Bph68S, Liangyou8676 and RHT presents excellent agronomic characteristic against BPH in rice field.

Comment 6: Another thing I didn't exactly understand from the presentation is the significance of female nymph versus male nymphs? What are the prior findings in the literature about this observation in relevance to antibiosis? Why is it important? More clarification would be welcome.

Response 6: Because the BPH belongs to r-strategist pest, has the characteristic of high fecundity.

So it is the significance of female nymph versus male nymphs due to the more destructive of females are to rice.

In our prior research in field, we found that the number of BPH settled on rice, which carrying resistance genes were significantly less than those don't.  

Comment 7: The authors mentioned that at least one of the genes being tested as a source of resistance was isolated. A short review of what the proposed genetic mode of action for these resistance genes being tested would be welcome.

Response 7: We're also interested in genetic mode of action for these resistance genes, but it  have not been tested. The rice lines acquired the resistance genes were obtained via crossing, backcrossing and molecular marker-assisted selection (MAS). We're going to try to explore genetic mode of action for these resistance genes in future.

Comment 8: For the writing style- I have noticed some interesting uses of "meanwhile" in the text. I would suggest that the authors review the use of the word in the text.

Response 8: We used “in the meantime” instead of “meanwhile” in the revised manuscript. Please see the revised manuscript for details.

Comment 9: Finally- as the authors discovered that the Bph3 may have a stronger antibiosis effect on BPH then Bph14/15 stack- I would have liked to see this spelled out and maybe a short discussion of what they plan to do next- Future work?

Response 9: We have added the relevant discussion of what we plan to do in future. Please see the revised manuscript for details.
